# Intraoperative Computed Tomography-Based Navigation with Augmented Reality for Lateral Approaches to the Spine

**DOI:** 10.3390/brainsci11050646

**Published:** 2021-05-15

**Authors:** Mirza Pojskić, Miriam Bopp, Benjamin Saß, Andreas Kirschbaum, Christopher Nimsky, Barbara Carl

**Affiliations:** 1Department of Neurosurgery, University of Marburg, Baldingerstraße, 35043 Marburg, Germany; bauermi@med.uni-marburg.de (M.B.); sassb@med.uni-marburg.de (B.S.); nimsky@med.uni-marburg.de (C.N.); Barbara.carl@helios-gesundheit.de (B.C.); 2Marburg Center for Mind, Brain and Behavior (MCMBB), 35043 Marburg, Germany; 3Department of Visceral, Thoracic and Vascular Surgery, University of Marburg, 35043 Marburg, Germany; akirschb@med.uni-marburg.de; 4Department of Neurosurgery, Helios Dr. Horst Schmidt Kliniken, 65199 Wiesbaden, Germany

**Keywords:** augmented reality, computer-assisted surgery, effective radiation dose, image-guided surgery, intraoperative imaging, spine navigation, lateral approach to the spine

## Abstract

**Background.** Lateral approaches to the spine have gained increased popularity due to enabling minimally invasive access to the spine, less blood loss, decreased operative time, and less postoperative pain. The objective of the study was to analyze the use of intraoperative computed tomography with navigation and the implementation of augmented reality in facilitating a lateral approach to the spine. **Methods.** We prospectively analyzed all patients who underwent surgery with a lateral approach to the spine from September 2016 to January 2021 using intraoperative CT applying a 32-slice movable CT scanner, which was used for automatic navigation registration. Sixteen patients, with a median age of 64.3 years, were operated on using a lateral approach to the thoracic and lumbar spine and using intraoperative CT with navigation. Indications included a herniated disc (six patients), tumors (seven), instability following the fracture of the thoracic or lumbar vertebra (two), and spondylodiscitis (one). **Results.** Automatic registration, applying intraoperative CT, resulted in high accuracy (target registration error: 0.84 ± 0.10 mm). The effective radiation dose of the registration CT scans was 6.16 ± 3.91 mSv. In seven patients, a control iCT scan was performed for resection and implant control, with an ED of 4.51 ± 2.48 mSv. Augmented reality (AR) was used to support surgery in 11 cases, by visualizing the tumor outline, pedicle screws, herniated discs, and surrounding structures. Of the 16 patients, corpectomy was performed in six patients with the implantation of an expandable cage, and one patient underwent discectomy using the XLIF technique. One patient experienced perioperative complications. One patient died in the early postoperative course due to severe cardiorespiratory failure. Ten patients had improved and five had unchanged neurological status at the 3-month follow up. **Conclusions.** Intraoperative computed tomography with navigation facilitates the application of lateral approaches to the spine for a variety of indications, including fusion procedures, tumor resection, and herniated disc surgery.

## 1. Introduction

In recent years, the lateral approach to the thoracic and lumbar spine has become one of the standard methods for the achievement of fusion [1], also known as extreme lateral interbody fusion (XLIF). The XLIF technique for the lumbar spine was initially described by Ozgur et al. [2], and it has also found application in pathology of the thoracic spine [3]. Lateral interbody fusion has been shown to be efficient for neuroforaminal stenosis decompression [4] and in the treatment of disc herniation, fracture, tumor, pseudoarthrosis, and junctional kyphosis [5]. XLIF has established itself as a useful method in revision surgery because it allows valid arthrodesis, avoiding the scar tissue created by previous posterior approaches [6].

Several studies have presented the use of iCT and spinal navigation for lateral approaches to the spine in the surgery of herniated discs and thoracic burst fractures, and for lateral interbody fusion [7,8,9,10,11,12,13]. In this paper, we report on our experience with 16 patients who underwent surgery via a lateral approach for herniated discs, degenerative spine disease with instability, primary and metastatic spinal tumors, and spondylodiscitis. To our knowledge, this is the first study which describes the use of iCT guided spinal navigation with the use of augmented reality (AR) for the lateral approach to oncological and infectious diseases of the spine.

## 2. Materials and Methods

We prospectively analyzed patients who underwent surgery with a lateral approach to the spine from September 2016 to January 2021. There were total of 104 surgeries performed at our center with a lateral approach to the spine. Of this number, 16 patients (15.4%) were operated on using iCT. Eighty-eight patients operated on via the lateral approach were not navigated. Of the patients who were non-navigated, 65 patients underwent XLIF, one patient underwent a resection of a giant ganglioneuroma of the thoracic spine (combined surgery with thoracic surgeon), and 22 patients underwent corpectomy with the implantation of an expandable vertebral body cage for various indications, including fracture, metastasis, and infection (14 in the lumbar spine and eight in the thoracic spine). Among these, 16 patients were investigated by intraoperative CT, applying a 32-slice movable CT scanner, which was used for automatic navigation registration. Inclusion criteria in this study were all cases which were operated on using iCT-based navigation with AR for lateral approaches to the spine. iCT was used when judged necessary by the operating surgeons as an important tool for orientation in the operative field and for control of the extent of resection or position of implants, especially in cases of revision surgeries using the same approach.

For oncological diseases, indications included cases of large tumors with invasion of the retroperitoneal space (Patients 4, 9, and 14), one patient with a giant cell tumor of Th12 (Patient 12), and patients with spinal instability due to pathological fracture of the vertebra due to metastasis (Patient 7).

For degenerative diseases, iCT was performed in patients with calcified herniated discs of the thoracic spine, where navigation facilitated the approach and where the extent of resection of the disc could be confirmed during the procedure with iCT when needed (Patients 3, 8, 10, 13, 15, and 16). Patients 1, 5, and 6 were cases of spinal instability, in which corpectomy and implantation of an expandable vertebral body cage was performed as revision surgery, so the application of iCT-navigation for the correct positioning of implants was considered crucial.

One case of spondylodiscitis (Patient 2) underwent XLIF from the right side. The indication for iCT and navigation in this particular case was the fact that this was not the standard approach, because XLIF is performed via the left transpsoas approach in our institution. Patient data and clinical results are summarized in Table 1. Informed consent was obtained from all individual participants included in the study. We obtained approval from the local ethics committee for prospective archiving of clinical and technical data for applying intraoperative imaging and navigation.

Standard C-arm X-ray was used prior to skin incision for level definition. A movable 32-slice CT-scanner (AIRO, Brainlab, Munich, Germany) was used for intraoperative CT (iCT). For iCT no patient movement was necessary. Details describing the setup were previously published, and no major modifications were necessary to apply the technique. [14,15,16,17].

Patients were positioned in the lateral decubitus position. Invasive electrophysiological monitoring, including electromyography (EMG) of the lumbar plexus, motor evoked potentials (MEPs), and somatic sensory evoked potentials (SSEPs), was routinely used for all cases. The reference array was attached either to an extra arm of the retractor system or on the iliac crest via a separate small incision. iCT was performed after exposing the spine, while the retractors were not removed. To enable automatic registration, scanner and patient were tracked during iCT. For registration scanning, dose-reduced protocols were used (helical acquisition, 33 mA) [15]. Registration accuracy was checked by placing the pointer tip on anatomical landmarks, such as the lateral surface of the vertebra body, or artificial landmarks, such as attached skin fiducials, retractor arms, reflecting spheres of the registration array, or clips placed in the surgical fields. The target registration error was measured by the offset of skin fiducials placed close to the incision, which were not part of the registration process.

For calculation of the effective dose (ED), the total dose length product (DLP) was multiplied by ED/DLP conversion factors, which were estimated to be 17.8 µ Sv/Gycm for thoracic and 19.8 µ Sv/Gycm for the lumbar spine [18,19]. The DLP refers to a phantom with a diameter of 32 cm for thoracic and lumbar examinations. In scans covering the thoracolumbar junction, the conversion factors were weighted according to the number of vertebrae covered. After a rough rigid prealignment, nonlinear registration of iCT data and preoperative image sets was performed (spine curvature correction element, Brainlab). Image fusion accuracy was carefully checked by inspecting the close matching of the outline of the preoperatively segmented vertebra in the iCT image. Repeat iCT scans, if any, were also scanned using the automatic registration setting. An initial bony navigation with a registration accuracy check was performed using intraoperative registration low dose iCT. Registration iCT is sufficient for bony navigation, however, we consider fusion with preoperative CT and MRI necessary for segmentation of anatomical structures, especially when AR has been applied (segmentation of herniated disc, tumor, vessels, vertebrae, etc.)

For AR support, the heads-up display of the operating microscope Kinevo900 (Zeiss, Oberkochen, Germany) was used. A registration array attached to the microscope allowed its position to be tracked. For controlling AR calibration, the microscope was centered above the patient reference array, so that the alignment of the AR visualization of the reference array and the optical information could be checked and adjusted if necessary. Various 3D objects can be visualized by AR, either in semitransparent, solid, or outlined fashion. AR registration accuracy was repeatedly ensured by focusing with the operating microscope on known structures, such as the edges of the retractor systems, and checking the position of the crosshair representation of the focus point in the AR visualization. The 3D objects representing the individual vertebrae could be switched on and off for each single vertebra and were generated applying the anatomical mapping software (Brainlab, Munich, Germany). After auto-segmentation, user interaction was needed to fine-tune the segmentation results using a smart brush feature, which was also used to segment the tumor extent. Implants were segmented by thresholding. Preoperative image data from CT and magnetic resonance imaging (MRI) were fused non-linearly by applying the spine curvature element (Brainlab, Munich, Germany). Preoperative CT and iCT data were used to segment the bony outlines of the vertebra, and co-registered MRI was mostly used to segment the tumor outline, due to its much better soft tissue contrast. For landmark checks to verify nonlinear fusion, the segmented vertebra outlines were visualized. The fused datasets were visualized in the spinal navigation application. The microscope application allows the visualization of the 3D objects in a semitransparent or solid mode superimposed on the microscope video; it displays probe’s-eye views, target views of the 3D objects, and a 3D overview depicting how the video frame is positioned in relation to the iCT data.

## 3. Results

### 3.1. General Characteristics of the Patients

Patient data and their clinical results are summarized in Table 1. One patient died in the early postoperative course due to cardiorespiratory failure (patient no. 12). One patient had several postoperative complications (patient no. 13), such as wound healing deficit, postoperative bleeding, pleural effusion, and pulmonary embolism, and one of these complications was related to the lateral approach (pleural effusion following transpleural surgery). Ten patients had improved, and five had unchanged, neurological status at the 3-month postoperative follow up.

### 3.2. Registration Accuracy

Automatic patient registration without user interaction resulted in high navigation accuracy, with a target mean registration error (TRE) of 0.84 ± 0.10 mm. Multimodal image data could be successfully fused nonlinearly with the iCT registration scan. Figure 1 demonstrates methods of checking the registration accuracy. In patients 9, 11, and 13, the reference frame was placed on the iliac crest, and for all the other patients, on the retractor arm. We did not measure the distance from the reference frame to the surgical site; however, for cases where the reference frame was attached to the retractor arm, we estimate this distance to be between 15 and 25 cm. We did not any find differences in accuracy between the placement of the reference array at the iliac crest or the retractor.

### 3.3. The Effective Radiation Dose

The effective radiation dose of the registration CT scans was 6.16 ± 3.91, whereas the scan range of iCT was defined by the surgical exposure of the spine. Table 2 summarizes scan length and DLP of the scout and registration scan and total effective dose, in addition to fused image sets and visualized objects. In patients 3, 5, 7, 8, 11, 12, and 15 a control scan for implant control and extent of resection control was performed. Table 3 summarizes the scan length and DLP of the scout and control scan and the total effective dose. In cases in which implants were inserted (patients 5, 7, 8, 11, and 12), and particularly vertebral body replacements (patients 5, 7, 11, and 12), repeat iCT scanning to check the implant position could be used to update the navigation. ED for the control scan was 4.51 ± 2.48 mSv. Segmentation of the implant and its visualization with updated AR allowed the checking of patient accuracy and patient registration, image registration, and AR calibration [20]. In this series, no revision surgeries were needed following control iCT. The total patient exposure, i.e., cumulative effective radiation dose for patients who received control iCT scan, was as follows: patient 3, 10.44 mSv; patient 5, 18.06 mSV; patient 7, 6.8 mSv; patient 8, 5.81 mSV; patient 11, 6.39 mSv; patient 12, 7.66 mSV; and patient 15, 24.19 mSV. Total patient exposure for these seven patients was 11.33 mSv ± 6.55.

### 3.4. Augmented Reality

Augmented reality was integrated into the surgical workflow for 10 of the 16 patients without problems (patients 7–16, Figure 2, Figure 3, Figure 4, Figure 5, Figure 6, Figure 7, Figure 8 and Figure 9). In cases of tumor resection (patients 4, 7, 9, and 11) and herniated disc (patients 10 and 16), a postoperative MRI for resection control was performed (Figure 4 and Figure 7). The overall AR accuracy was ensured during the procedure by focusing on the center of the skin fiducial and checking the close overlay of the AR representation of the reference array and reality. iCT scanning and automatic AR registration required an additional intraoperative time of less than 5 min. The HUD was turned on or off at the request of the surgeon, to allow optional display of objects and prevent distraction from the operative field due to too much information. AR-supported surgery improved surgeon comfort and led to better understanding of the 3D anatomy, both of which are prerequisites to avoid severe complications.

### 3.5. Clinical Application of iCT and AR

Patient 1 had adjacent segment disease L1/2 following lumbosacral spondylodesis and implantation of a vertebral body cage, following L2 corpectomy via lateral approach in another institution. Dislocation of the L2 cage with adjacent segment disease and fracture of L1 occurred. For this revision surgery, using the left lateral transpsoas approach, iCT-based navigation facilitated the approach through the scar tissue up to the vertebral body implant. From the outline of the implant, the navigation accuracy was able to be checked. Following removal of the implant and using navigation, the extent of resection of L1 and L2 was able to be correctly assessed and the vertebral body implant replaced.

Patient 2 had spondylodiscitis L4/5 with a large abscess of the psoas muscle on the right side. We decided to drain the psoas abscess and perform a nucleotomy and XLIF of an infected segmented. As we did not use the standard left approach, iCT-based navigation was used, which facilitated the approach and XLIF procedure.

Patient 3 had a calcified herniated disc Th9/10. Initial localization and incision were determined using X-ray. Registration scan was performed following placement of the transpleural retractor, with the reference frame attached to the retractor. Using iCT-based navigation and AR, one-third of the posterior portion of the vertebral body T9 and 10 with calcified disc was resected, starting the resection using a diamond drill from the posterior aspect of the vertebra towards the spinal canal. The herniated disc was differentiated from surrounding neural tissue by a different color, which allowed determination of the optimal trajectory to reach the pathology. Once the dural sac was exposed, micro-instruments were used for the careful preparation of the herniated disc from the spinal cord. Navigation was updated on the bony landmarks, and guided resection throughout the procedure was performed until the disc was completely resected. Control iCT confirmed the complete extent of resection.

Patient 4 had a giant aneurysmatic bone cyst with origin on the 8th rib, and which had invaded the neuroforamen Th8/9, with partial destruction of Th8 and Th9. We decided to perform hemilaminectomy and release the nerves Th8 and 9 in their foramina from the tumor with nerve root resection. Following this, the patient underwent combined surgery via a lateral approach in conjunction with a thoracic surgeon. A registration scan was performed following placement of the retractor. Partial rib resection was performed and complete resection of the intrathoracic part of the tumor up to exposure of the laminectomy defect and the dural sac from the lateral approach. iCT navigation enabled safe resection throughout the procedure, with resection of tumor remnants on the costotransverse joints. Control iCT confirmed the desired total amount of resection.

Patient 5 underwent navigated corpectomy of L2 following instability due to fracture. A registration scan was performed following placement of the retractor, and the bony outlines of the vertebra and the adjacent discs were used for navigated corpectomy and placement of the vertebral body implant.

Patient 6 developed instability following kyphoplasty of Th12. Following dorsal stabilization, navigated corpectomy of previously cemented vertebra was performed with iCT-based navigation following a registration scan after placement of the retractor. The correct position of the vertebral body implant was verified in the control scan.

Patient 7 was the first case in which AR was used in this series. The patient had a pathological L2 fracture due to metastasis of breast cancer. Following dorsal stabilization, lateral approach for corpectomy was performed. A registration scan was performed following placement of the retractor with the reference frame placed on the retractor. The bone substance was very soft and the borders to adjacent segments were unclear due to metastasis. Outlines of the vertebra were shown in the microscope-based AR, which enabled the desired resection. The control scan showed the correct position of the vertebral body implant. Use of iCT and AR enabled 3D visualization of the implant and adjacent vertebra.

Patient 8 had a calcified herniated disc TH8/9 with paraparesis. Initially, dorsal surgery with partial resection of the herniated disc was performed. Postoperative MRI revealed a disc remnant with compression of the spinal cord, so the decision was made to perform a navigated resection of the remaining disc via lateral approach. Registration scan was performed following placement of the retractor. A segmented outline of the herniated disc remnant and adjacent vertebra was shown in the microscope-based AR in the overlay fashion, which directed the lateral decompression to the herniated disc and led to its excision.

Patient 9 had a giant retroperitoneal neurinoma, with origin on the L2 nerve (Figure 5 and Figure 6). The reference frame was attached to the pelvic crest and the registration scan performed following the placement of the retractor. Microscope-based AR with iCT navigation was particularly helpful for excision of the tumor parts bordering the vertebra and aorta following tumor debulking. The tumor was a solid, partially calcified mass, so there was no significant positional shift following debulking. The visualization of the tumor outline in the AR is of use, even when the registration accuracy is compromised, because the size of the object is still displayed correctly and the extent of the tumor can be estimated with certainty. Gross total resection was achieved.

Patient 10 had a herniated disc Th7/8 (Figure 2, Figure 3 and Figure 4). A registration scan was performed following placement of the retractor. In the microscope, the 3D outlines of the vertebra and the segmented disc herniation (in blue) were visualized by the HUD, which largely facilitated the orientation and resection.

Patient 11 had a giant cell tumor of the Th12. Following dorsal stabilization, corpectomy with tumor resection was performed via lateral approach. A registration scan was performed following placement of the retractor, and the reference frame was placed on the iliac crest. Navigation enabled a continuous overview of the desired extent of corpectomy. Control iCT showed the correct position of the implant. Fusion of the preoperative CT and MRI with control iCT showed the planned resection extent and the correct position of the implant.

Patient 12 had a pincer type A2 fracture of Th12. Lateral approach with partial rib resection was performed. A registration scan was performed following placement of the retractor, with the reference frame attached to the retractor. Using navigation and microscope-based AR, outlines of the fractured vertebra and its borders with the spinal canal were visualized, which facilitated the corpectomy and discectomy of adjacent vertebra. Following this, the vertebral body implant was placed in the corpectomy defect, and its correct position was confirmed in the control iCT scan. Additional dorsal stabilization of Th11-L1 for stability was subsequently performed.

Patient 13 had a large calcified herniated disc Th11/12 with myelopathy. We decided to perform a two-stage surgery. Initially, the patient underwent dorsal decompression with stabilization of Th10/11, with a non-navigated transpleural lateral approach for resection of the herniated disc. Follow-up CT and MRI showed incomplete resection, so a navigated resection with iCT and AR using transpleural lateral approach was performed. A registration scan was performed following the placement of the retractor, and the reference frame was placed at the iliac crest. A complete resection of the disc remnant was performed.

Patient 14 had a Th11/12 schwannoma with a large retropleural portion of the tumor. Resection of the T12 nerve root via dorsal approach was performed following lateral approach for resection of the retropleural tumor portion. A registration scan was performed following the placement of the retractor, and the reference frame was placed on the retractor arm. iCT-based navigation and microscope-based AR with outlines of the tumor, vertebra, and aorta, which were shown in overlay fashion in the operative microscope, facilitated the orientation in the surgical field, particularly during the resection of the tumor portions adjacent to the aorta. Follow-up MRI confirmed the complete resection of the tumor.

Patient 15 had a large calcified herniated disc Th 7/8 and had undergone previous surgery in an external hospital with left sided hemilaminectomy Th7/8. Due to worsening of the paraparesis following the primary surgery, the patient was transferred to our department. Left lateral transpleural approach with costotransversectomy was performed with a registration scan following implantation of the retractor. The reference frame was placed on the retractor arm. Microscope-based AR with outlines of the vertebras and herniated disc, shown in overlay fashion in the microscope HUD, directed the dissection and facilitated orientation in the surgical field. Complete resection of the herniated disc was performed, which was confirmed in the control iCT scan (Figure 8 and Figure 9, Appendix A: microsurgical resection of herniated calficied disc video).

Patient 16 had a large calcified herniated disc Th9/10 with myelopathy and ataxia. A left lateral retropleural approach with costotransversectomy was performed. A registration scan was performed following placement of the retractor, and the reference frame was attached to the retractor arm. Microscope-based AR with outlines of the vertebrae and herniated discs, shown in overlay fashion in the microscope HUD, facilitated the orientation because the borders between the calcified disc, adjacent vertebrae, and the spinal canal were not clear. Partial posterior one-third corpectomy of Th9 and 10 was performed with a diamond drill following subtotal resection of the calcified disc, with meticulous dissection of disc remnants adjacent to the spinal cord. Follow-up CT and MRI confirmed the desired extent of resection.

Eighty-eight patients operated on via lateral approach during the same period at our institution were not navigated. If we exclude 65 XLIF cases, one patient underwent a resection of a giant ganglioneuroma of the thoracic spine and 22 patients underwent corpectomy with implantation of an expandable body cage for various indications, including fracture, metastasis, and infection (14 in lumbar spine and 8 in the thoracic spine). The patient who underwent a resection of a ganglioglioma had a partial resection that possibly could have been avoided if iCT-based navigation and AR had been used. From patients who underwent corpectomy and implantation of vertebral body implants during the same period, three patients underwent revision surgery due to dislocation of the implant and adjacent segment disease; which can be compared to the navigated cases (patients 1, 2, 5, 6, and 7), where no revision surgeries were needed. Since patients in both groups were non-homogenous regarding pathology and different operative techniques, we considered a comparison between the two groups impractical.

## 4. Discussion

The navigated implantation of pedicle screws uses a single vertebra registration based on surface matching. This is due to the direct identification of the bony surface by sampling at least 20 points on the lamina of a vertebra with the tip of navigation pointer, resulting in a low registration error of 1.1 ± 0.61 mm in phantom studies [21]. However, this strategy is not applicable to the lateral approach to the spine because it can be implemented only in posterior open surgery. In spine procedures, fiducial-based registration is an option only when intraoperative imaging is available, because flexibility of the spine precludes the vertebra alignment during the preoperative scan being identical to the intraoperative alignment [20,22]. User-independent registration can be achieved by intraoperative imaging, which, regarding spinal surgery, has the additional advantage that the positioning of the patient and vertebral alignment during imaging correspond to the real intraoperative situation. A user-independent automatic registration concept applying intraoperative CT was recently reported and showed a mean target registration error (TRE) for spinal procedures of 0.86 ± 0.28 mm [20]. Our results correspond well to these and to results of the spine phantom studies comparing intraoperative registration to bone surface registration or to bone-implanted miniscrews, which showed TREs of 0.74 and 0.14–0.78 mm, respectively [23,24]. Radiation-free alternatives for registration are surface-matching and point-to-point registration using a pointer or settings using ultrasound to delineate the shape of the bony structures of the spine for registration; however, these are considerably more inaccurate than automatic iCT-based registration [17,25,26]. We have already published our initial experience with iCT-based navigation with AR for spine surgery [13,16,17], with only two patients who underwent surgery via a lateral approach. We aimed to present our experience on the use of this method in a lateral approach for spine surgery for various indications.

Several studies have been published describing the use of iCT for lateral approaches to the spine [7,8,9,10,11,12]. iCT using navigation assistance with electrophysiological monitoring was reported for lateral lumbar discectomy [8,11] for thoracic burst fractures [12] and cage placement in lateral lumbar interbody fusion procedures [7,9,10]. iCT was used for precise localization within the exposure corridor for lateral retroperitoneal transforaminal approaches for large L1/2 disc herniations, limiting the amount of bony resection and dissection through the psoas muscle [11], and for accurate cage placement [7]. Navigation was used to confirm the amount of decompression to the contralateral pedicle with a significant reduction in the number of fluoroscopic localization images [11]. When indirect decompression with implantation of the cage was not able to be shown in the iCT, further posterior decompression was performed [9]. Control iCTs in this study were performed for resection control in cases of herniated discs, vertebral tumors, and neurinoma, and for implant control. This enabled direct quality control, and showed that the aim of surgery was achieved, with the desired amount of resection being accomplished and the implants positioned correctly. Implant segmentation with AR update allows the checking of patient accuracy, registration, image registration, and AR calibration [20].

As the user is positioned behind protective shielding during three-dimensional imaging, exposure is minimized, as the navigation is radiation-free [27]. The effective radiation dose of the registration CT scans was 5.76 ± 3.32 mSv. The mean ED for initial registration scans applying low-dose protocols in a recent study was 4.12 ± 2.13 mSv (1.48–9.64 mSV) for the thoracic region, and 3.37 ± 0.93 mSv (1.59–5.01 mSv) for the lumbar region [20]. In relation to standard diagnostic scans, this was a 2.5 to 6-fold reduction for spinal scans compared to standard diagnostic scans [28,29]. A recent study of patients who underwent pedicle screw placement showed an average patient ED of 15.8 ± 1.8 mSv, which mainly correlated with the number of vertebrae treated and the number of cone-beam computed tomography acquisitions performed [27]. Another study reported an average total radiation exposure of 5.69 mSv for the patient, whereas thoracic and lumbar instrumentations had higher radiation emission than the cervical, deformity, and degenerative cases, which caused more emission than oncology or trauma cases [30]. The radiation exposure for the surgeon was reported to be significantly lower in all the studies when using iCT for lateral approaches to the spine [7,8,9,10,11]. iCT with integrated navigation systems in spinal stabilization has been shown to be rapid and easy to perform, without restricting access to the patient, and, by replacing pre- and post-operative imaging, is not associated with an additional exposure to radiation [31]. None of the patients who received instrumentation in our study experienced implant failure at follow up. A recent review of lumbar interbody fusion in patients with osteopenia and osteoporosis revealed that Hounsfield units may be an effective one-tool arsenal when evaluating these patients preoperatively with respect to the loosening or dislocation of implants [32].

One patient in our study experienced an approach-related complication, with pleural effusion following the transpleural approach to the thoracic spine. Complications related to this approach have been described to occur more frequently and range up to 25% [7]. The reported major morbidity and mortality rates of XLIF have led to an argument about whether it should remain part of the spinal instrumentation [33]; however, the procedure has been shown to be safe with a relatively low complication rate when performed by experienced surgeons [34], and from which the postoperative deficits are transient in their nature [35] and reveal a lower incidence of infection [36]. One of the most common complications that is resolved without any intervention in the transpsoas approach is lumbar plexus injury, which can lead to anterior thigh pain, sensory changes, and weakness in hip flexors. Preoperative psoas major muscle volume was not found to be correlated with postoperative anterior thigh numbness, pain, or weakness in a recent study [37]. Further studies, focusing on the muscle morphology or fatty infiltration, and to reduce the complication rates associated with this procedure are needed [37], while iCT with segmentation of the muscle in AR could provide additional information on a possible connection between its morphology and complication rate.

The use of AR in degenerative and oncological spine surgery has recently been described in conjunction with its potential for application in complex anatomical situations and for resident education [13,16,17]. The previously described advantages of the use of AR have also been shown here [13,16], particularly in tumor surgery for identification of tumor extent by visualizing the tumor outline, and for visualization of cutting lines for corpectomy and extirpation of the calcified herniated disc.

AR systems have demonstrated a high accuracy compared to free-hand and conventional navigation in spine surgery in several clinical studies [38]. Good 3D depth perception, resulting in smooth hand–eye coordination, was provided using advanced AR visualization in combination with additional views on screens near the surgical field [39]. To ensure AR can support anatomical orientation and visualize objects in the surgical field itself, its correct registration—which depends on correct nonlinear image registration, correct calibration of the HUD of the operating microscope, and correct patient registration—is crucial [13]. AR has been found to increase the comfort for the surgeon in reoperations, in which anatomical landmarks are lacking [16]. Furthermore, its use as a tool for resident education should not be underestimated [16]. Visualization of tumor outline in the AR is of use, even when registration accuracy is compromised, because the size of an object is still displayed correctly and the extent of the tumor can be estimated with certainty [16]. AR has been applied to spine surgery for pedicle screw placement, targeted cervical foraminotomy, bone biopsy, osteotomy planning, and percutaneous intervention [40]. When performing corpectomy and en bloc tumor resections, AR has been applied not only in open procedures, but also in stereotactic surgical navigation, as described in a recent case report on the resection of an L1 chordoma through a posterior-only approach, with visualization of the navigation aligned in parallel with the tracked instrument, providing maximum precision and safety [41]. The Augmented-Reality Head-Mounted Display Stereotactic Navigation System for Spine Surgery (AR-HMD) is a novel AR-based method that has demonstrated high precision and accuracy in the placement of pedicle screws [42]. Future goals include achieving maximal accuracy with a robotic arm and AR tracking of surgical tools [43,44].

AR aims to overlay virtual bony structures on patients in the setting of hybrid operating rooms that enable real-time feedback of all instruments in space and in relation to anatomical structures [43]. AR allows for the identification of the parts of the vertebra (pars, pedicle, and disc) when they are not in the exact field of view, which enables the surgeon to more safely and confidently perform maneuvers for resection of tissue or bone [45]. Herniated discs and synovial cysts can be clearly differentiated from surrounding neural tissue and bony landmarks by color, allowing the optimal trajectory to reach the pathology to be determined [45].

As nerve injury during the transpsoas approach is the most common and potentially most devastating complication of the XLIF procedure, several studies have looked at defining “safe working zones” for the placement of the retractor [1,35,46,47,48]. These studies have shown that when approaching the lumbar spine from L3, L2, or L1, the psoas muscle should be split into the ventral three-quarters of the vertebral body to avoid nerve injury [49]. Since vascular and lumbar plexus nerve injuries are a major cause of morbidity in the lateral approach to the spine [33], visualization of these structures is of vital importance for increasing the safety of the procedure.

Intraoperative imaging enables fewer registration errors and allows accurate information to be obtained regarding 3D spine configuration. Use of unsophisticated nonlinear image registration approaches for compensation of spatial flexibility may lead to errors due to spinal flexibility [20]. In these cases, the importance of a low number of registration errors justifies the use of intraoperative imaging and, in particular, the radiation exposure for the patient. Limitations of our study are the small number of patients and the short follow-up times. A further limitation of our study is the non-homogeneity of the small sample, due to the inclusion of multiple pathologies (infectious, oncological, and degenerative diseases); however, in our experience, iCT with AR for the lateral approach has been shown to be suitable for all of the included pathologies. Although we present a limited number of cases, intraoperative imaging provided immediate intraoperative quality control, thus avoiding complications and allowing the evaluation of whether the aim of the surgery was met, i.e., providing information on the extent of tumor resection, or resection of the herniated disc, or whether implants were correctly placed according to the plan. A control group with non-iCT-navigated cases for lateral approaches to the spine could help provide an objective assessment of the use of iCT-based navigation and AR; however, we considered this to be impractical in a prospective setting because this would require not using iCT and AR for patients for whom the surgeon assessed that these techniques would increase patient safety and surgeon comfort.

A future possible upgrade to the use of intraoperative CT in lateral approaches to the spine is to define “safe-working zones”, with the segmentation of the lumbar plexus in the navigation workflow, thereby providing additional safety for the procedure in combination with neuromonitoring.

## 5. Conclusions

Intraoperative computed tomography with navigation and the implementation of augmented reality facilitates the application of lateral approaches to the spine for a variety of indications, including fusion procedures, tumor resection, spondylodiscitis, and herniated disc surgery. Intraoperative imaging provided immediate intraoperative quality control, thus avoiding complications and allowing evaluation of whether the aim of the surgery was met.

## Figures and Tables

**Figure 1 brainsci-11-00646-f001:**
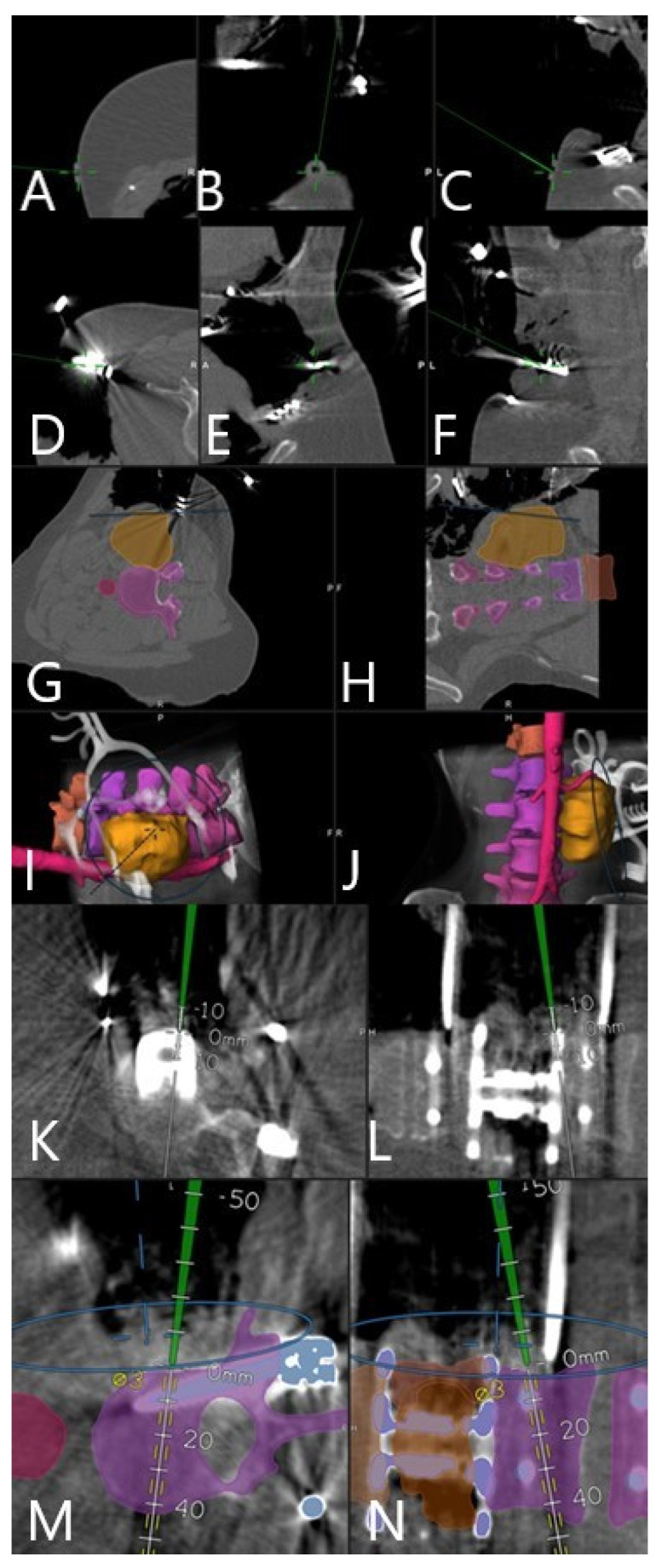
Navigation accuracy check using tip of the navigation pointer. (**A**–**C**) In the divot of a skin fiducial (patient no. 10). (**D**–**F**) On the spine retractor (patient no. 10). (**G**–**J**) On the outer tumor surface (patient no. 10). (**K**,**L**) On the expandable vertebral body cage (patient no. 11). (**M**,**N**) On the lateral surface of the vertebra (patient no. 11).

**Figure 2 brainsci-11-00646-f002:**
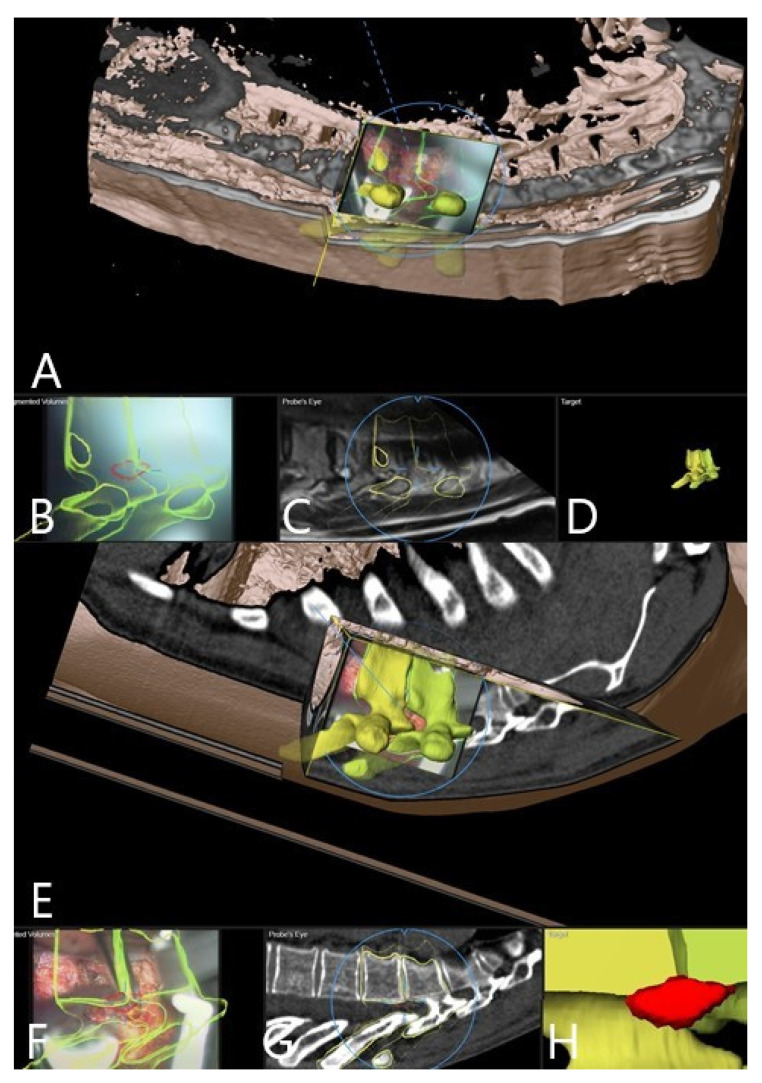
A 62 year old female patient (patient no.10) with herniated thoracic disc Th 7/8, operation via a left lateral transpleural approach. (**A**) Overview visualization depicting the position of the microscope view in relation to the segmented vertebra, visualized in yellow color. (**B**) AR visualization with the outline of vertebra bodies T7 and T8 and herniated disc in red. (**C**) Probe’s-eye view with segmented structures in the MRI. (**D**) Segmented objects visualized separately in the target view. (**E**) Overview visualization depicting the microscope view position related to segmented vertebra. (**F**) AR visualization with the outline of vertebra bodies T7 and T8 and herniated disc in red after placement of spinal retractor. (**G**) Probe’s-eye view with segmented structures in the intraoperative CT. (**H**) Target view.

**Figure 3 brainsci-11-00646-f003:**
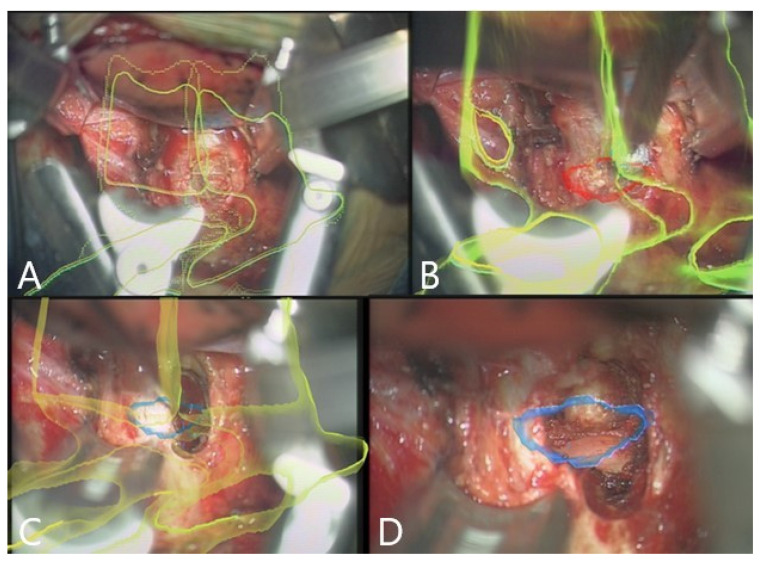
AR microscope view during the course of the surgery (same patient as in Figure 2). In yellow, the 3D outlines of the vertebra and in blue the segmented disc herniation are visualized by the HUD. (**A**) After retractor placement. (**B**) Removal of the sequester with a rongeur. (**C**) Mobilization of the disc fragment with the hook. (**D**) Following removal of the sequester, visualization of dura with outline of the extirpated disc.

**Figure 4 brainsci-11-00646-f004:**
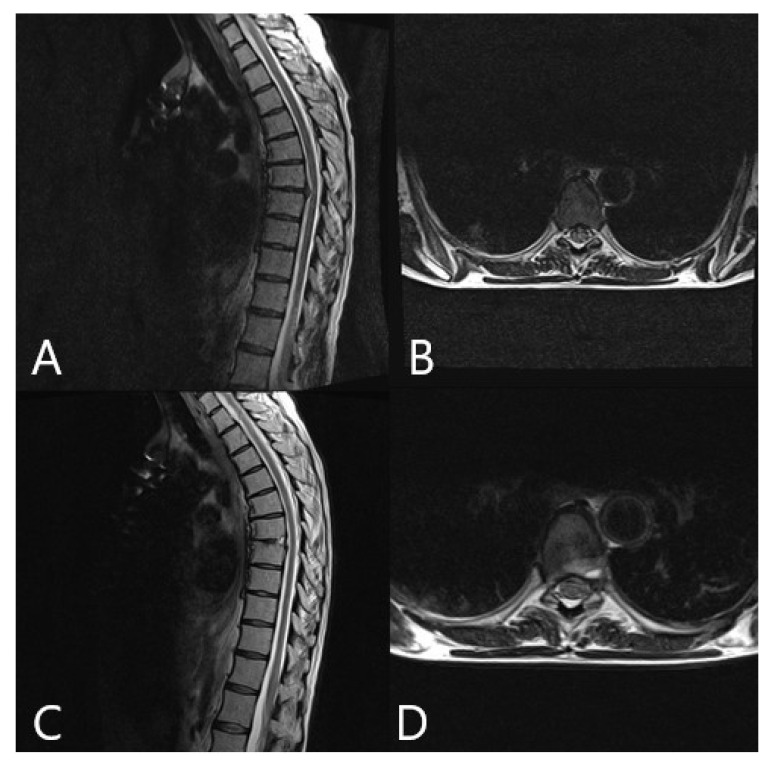
Same patient as in Figure 2 and Figure 3. (patient no. 10), surgery via a left transpleural approach. MRI of the thoracic spine. (**A**) Preoperative T2 sagittal. (**B**) Preoperative T2 axial. (**C**) Postoperative T2 sagittal, which shows complete resection of the herniated disc. (**D**) Postoperative T2 axial.

**Figure 5 brainsci-11-00646-f005:**
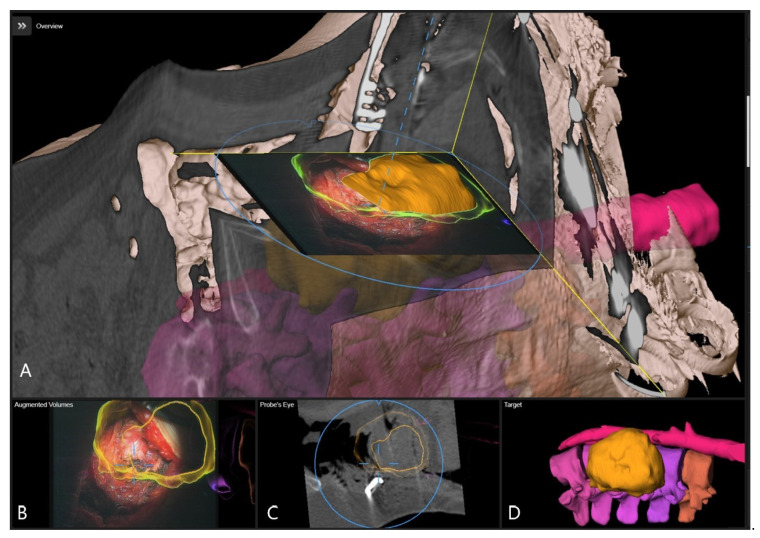
A 52 year-old female patient (patient no. 9) with a L2 neurinoma operated via a left lateral retroperitoneal approach. Registration accuracy check is shown in Figure 1. (**A**) Overview visualization depicting the position of the microscope view in relation to the segmented tumor (orange), vertebra (violet), and aorta (purple). (**B**) Microscope video, AR visualization with the outline of the tumor in orange. (**C**) Probe’s-eye view with segmented structures in the iCT. (**D**) Segmented objects visualized separately in target view.

**Figure 6 brainsci-11-00646-f006:**
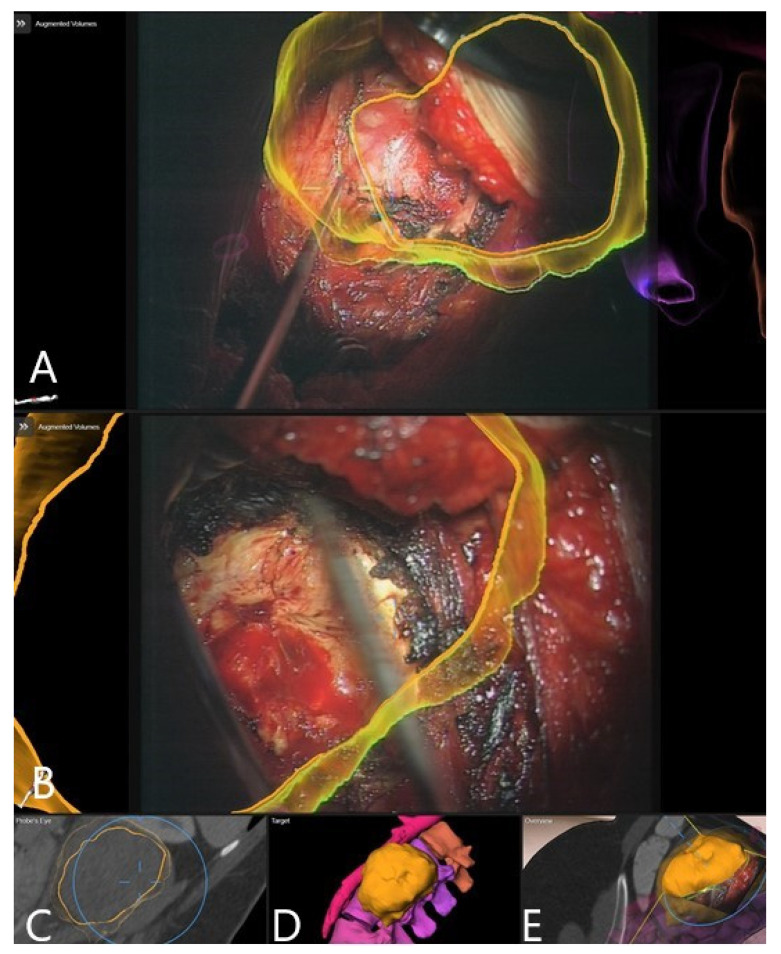
Same patient as in Figure 5 (patient no. 9). Microscope-based AR visualizing the tumor outline. (**A**) At the beginning of the tumor resection. (**B**) During the course of resection, with (**C**) Probe’s-eye view; (**D**) target view visualizing the displayed AR objects. (**E**) A 3D rendering of the iCT images illustrating how video frame is placed in relation to the image data.

**Figure 7 brainsci-11-00646-f007:**
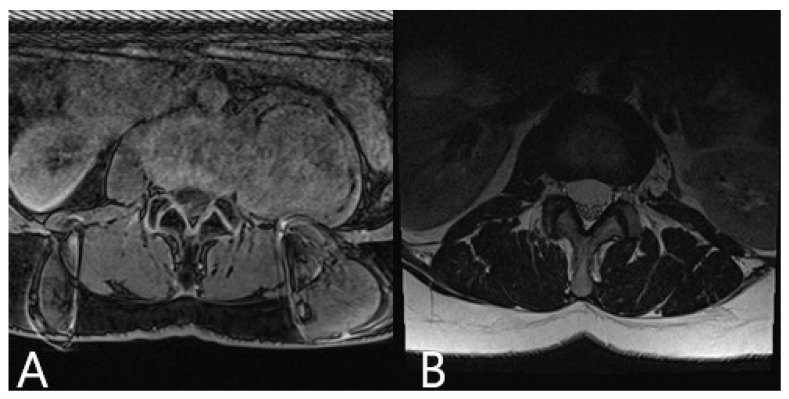
Same patient as in Figure 5 (patient no. 9). MRI of the lumbar spine. (**A**) Preoperative axial post-contrast MRI of the lumbar spine shows large left-sided retroperitoneal tumor with origin in the left L1/2 neuroforamen. (**B**) Postoperative axial T2 MRI of the lumbar spine shows complete resection of the neurinoma at the three-month follow up.

**Figure 8 brainsci-11-00646-f008:**
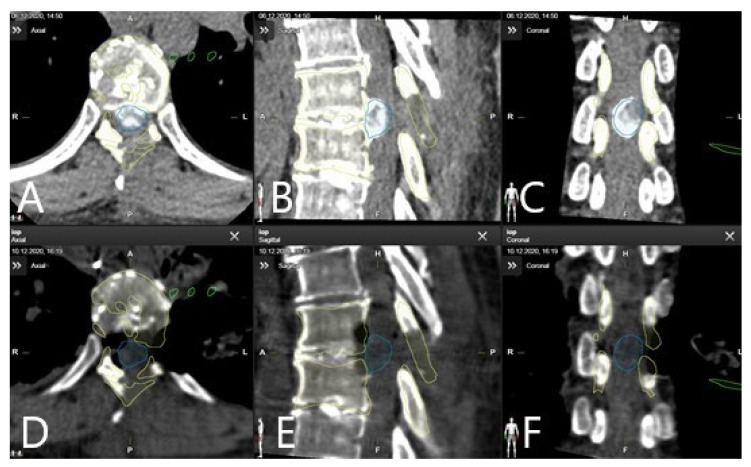
A 46-year-old patient with large calcified herniated disc Th 7/8, who underwent previous surgery in an external hospital with left-sided hemilaminectomy Th7/8 (patient no. 15). Due to worsening of the paraparesis following the primary surgery, patient was transferred to our department. Left lateral transpleural approach with costotransversectomy and complete resection of the herniated disc was performed. Intraoperative CT scan used for navigation registration performed following implantation of the retractor in (**A**) axial, (**B**) sagittal, and (**C**) coronal view, with segmented herniated disc (blue), vertebra Th7 and Th8 (yellow), and the XLIF–retractor (green). Control CT scan for extent of resection in (**D**) axial, (**E**) sagittal, and (**F**) coronal view shows complete resection of the herniated disc following costotransversectomy via a left lateral transpleural approach.

**Figure 9 brainsci-11-00646-f009:**
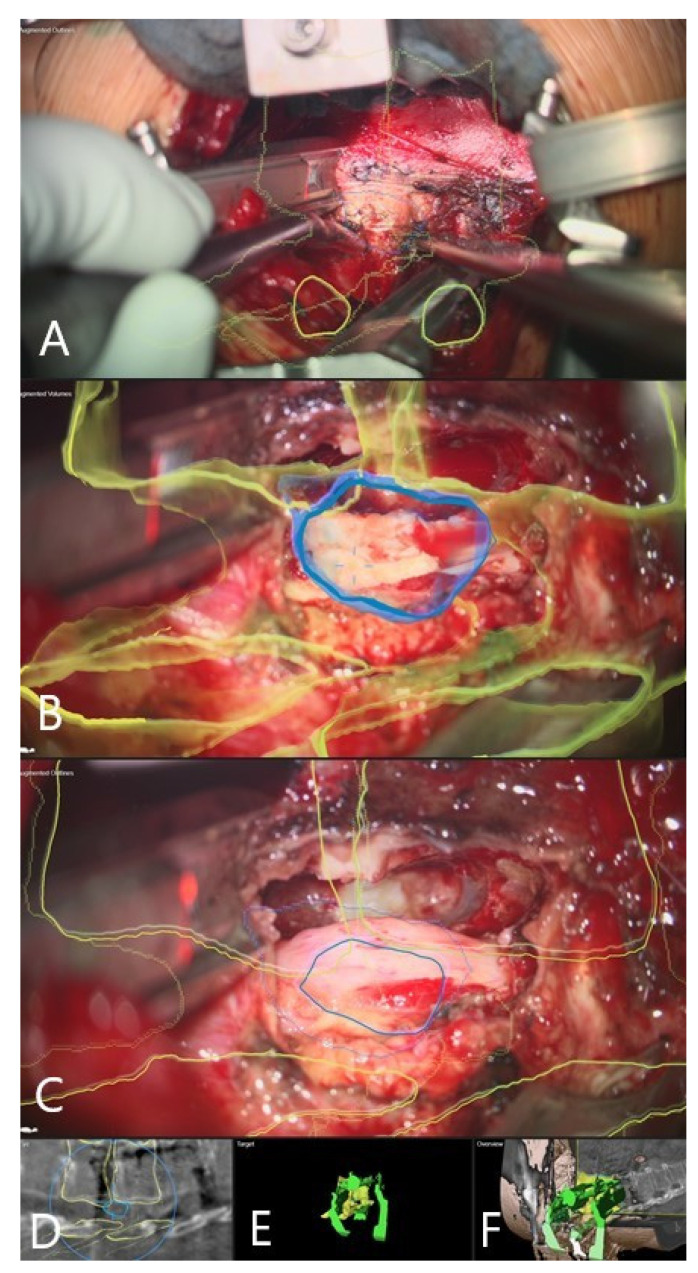
Same patient as in Figure 8 (patient no. 15). Microscope-based AR visualizing the outline of the herniated disc and vertebra Th 7 and 8. (**A**) At the beginning of the discectomy. (**B**) During the course of discectomy. (**C**) Following drilling out the sequestrated disc with exposure of the dural sack with (**D**) probe’s-eye view; (**E**) target view visualizing the displayed AR objects. (**F**) A 3D rendering of the iCT images, illustrating how video frame is placed in relation to the image data.

**Table 1 brainsci-11-00646-t001:** Patient characteristics and clinical results.

Number	Age	Sex	Diagnosis	Preoperative Symptoms	Procedure	Complications	Outcome
1	75	M	Adjacent segment disease L1/2 following spondylodesis L2-5 and implantation of expandable vertebral body cage at L2	Back pain, hip flexor paresis 4/5	1. Spondylodesis T9-S1 2. Removal of the L2 vertebral body implant, Corpectomy L1/2, Implantation of expandable vertebral body cage, left retroperitoneal approach	No	Regredient pain, no neurologic deficits following surgery
2	80	M	Spondylodiscitis L4/5 following surgery for right hip prothesis	Back pain	XLIF from right following Spondylodesis L4/5	No	Regredient pain, no neurologic deficits following surgery
3	66	F	Calcified herniated disc Th 9/10	Ataxia Paraparesis 4/5	Left lateral retropleural approach, sequestrectomy of the herniated disc	None	Improvement of ataxia following surgery
4	19	F	Giant aneurysmatic bone cyst Th 8/9	Back pain	1. Hemilaminectomy T8/9, Resection of the thoracic nerve origin 2. Tumor resection via left retropleural approach, Resection of 7/8 Rib with reconstruction of the thoracic wall	No	No pain, no deficits, and no recurrence at follow up
5	80	M	Instability following L1 vertebral body fracture and stabilization T11-L3	Back pain, Paraparesis 4/5	Corpectomy L1/2, Implantation of expandable vertebral body cage, left retroperitoneal approach	No	Improvement of pain, no deficits following surgery
6	77	M	Instability following kyphoplasty of T12 and stabilization T11-L1 due to T12 fracture	Back pain, Paraparesis 4/5	Corpectomy T12, Implantation of expandable vertebral cage, left retropleural approach	No	Improvement of pain, no deficits following surgery
7	F	76	L2 breast cancer metastasis	Back pain	1. L1-3 stabilization 2. Corpectomy L2, implantation of expandable vertebral cage, left retroperitoneal approach	No	Improvement of pain, no deficits following surgery
8	M	51	Herniated disc T8/9 with myelopathy	Back pain, paraparesis 3/5, urinary incontinence	1. Partial resection of the herniated disc via posterior approach with right costotransversectomy 2. Resection of the remaining disc via left retropleural approach 3. Stabilization T8-9	No	No pain and no deficits following surgery
9	F	52	L2 Neurinoma	Back and hip pain	Resection of neurinoma via left retroperitoneal approach	No	No pain, no deficits, and no tumor recurrence at follow up
10	F	63	Herniated disc T 7/8 with myelopathy	Worsening of back pain due to chronic pain syndrome following multiple spine surgeries, ataxia	Left lateral retropleural approach, sequestrectomy of the herniated disc	No	Chronic pain syndrome with moderate improvement, no ataxia, no deficits at follow up
11	F	51	Giant cell tumor of T12	Back pain, paraparesis 3/5, urinary incontinence	Left lateral retropleural approach, corpectomy T12, implantation of expandable cage	No	No pain, no deficits, and no tumor recurrence at follow up
12	F	71	T12 fracture	Back pain, paraparesis 3/5	1. Left lateral retropleural approach, corpectomy Th12, implantation of expandable cage 2. Th11-L1 stabilization	No	Death 5 weeks following surgery due to pneumonia, exacerbation of COPD and cardiorespiratory failure
13	F	50	Calcified herniated disc T10/11 with myelopathy and spinal canal stenosis	Back pain, paraparesis 2/5, urinary and stool incontinence	1. Dorsal stabilization T10-11 with spinal canal decompression	Pleural effusion in the field of the lateral operative approach, treated with thorax drainage	Improvement of back pain and paraparesis (4/5) with urinary incontinence, no stool incontinence 6 months following surgery
					2. Left lateral retropleural approach, partial resection of the herniated disc	Hematoma on 10th day following surgery with evacuation of hematoma in the dorsal operative field	
					3. Reoperation through left retropleural approach, placement of thorax drainage due to chambered pleural effusion and complete resection of the calcified disc	Dorsal wound revision due to healing deficit 4 weeks following surgery;	
14	F	48	Schwannoma Th11/12	Back pain	1. Resection of T12 nerve root via dorsal approach. 2. Left lateral transpleural approach, resection of the tumor	None	Improvement of back pain following surgery
15	F	46	Calcified herniated disc Th 7/8	Back, pain, paraparesis	Left lateral retropleural approach, constotransversectomy, total resection of the herniated disc	None	Improvement of back pain and paraparesis following surgery
16	M	38	Calcified herniated disc Th 9/10	Back pain, paraparesis	Left lateral retropleural approach, costotransversectomy, subtotal resection of the herniated disc	None	Improvement of back pain and paraparesis following surgery

**Table 2 brainsci-11-00646-t002:** Scan length and DLP of scout and registration scan, and total effective dose with fused image datasets with iCT registration and visualized objects in augmented reality.

Number	Protocol	Scout Scan DLP (mm)	Scout Scan Length (mm)	iCT Scan DLP (mGy.cm)	iCT Scan Length (mm)	Total DLP (mGy.cm)	Total ED (mSv)	Visualized Objects in Augmented Reality
1	L-spine	82.00944	265.9999	639.47850	126	721.48794	14.29	-
2	L-Spine 50%	34.18377	286	210.19560	94.999888	244.37937	4.84	-
3	T-Spine 30%	27.81074	223	242.90090	104	270.71164	4.82	-
4	T-spine 30%	21.53883	160.9999	315.9917	160	337.53053	6.01	-
5	L-spine 30%	18.20057	127.9999	438.00650	127	456.20707	9.03	-
6	T-Spine	107.99710	273	437.70600	93	545.7031	9.71	-
7	L-Spine 70%	12.34	192.00	167.04	167.00	179.38	3.55	vertebral body replacement, screws and rods, T12, L1, L2, L3, L4,
8	T-Spine 70%	35.59999	300.00	164.75950	188.00	200.36	3.57	spinal cord, spinal canal, C1, C2, C3, C4, C5, C6, C7, T1, T2, T3, T4,T5, T6, T7, T8, T9, T10, T11, T12, disc hernation
9	L-Spine 70%	13.80420	221.00	127.95180	168.00	141.76	2.81	tumor, T12,L1, L2, L3, L4, kidney, vessels
10	T-Spine 70%	51.66633	286.00	437.57950	248.00	489.25	8.71	T7, T8
11	T-Spine 70%	19.82	144.00	145.00	117.00	164.82	3.10	L3, L4, nerve root
12	T-Spine 70%	24.74	190	201.82	164	226.56000	4.03	Vertebral body T10, T11, T12, L1, L2, implant, clamp
13	T-Spine 90%	18.1	127	54.11	79	72.21000	1.29	Vertebral body T10,11, T10, T11, screws, disc herniation
14	T-Spine 70%	-	-	112.11	116	112.11	2.00	Vertebral body T10, T11, T12, L1, tumor, aorta, spinal cord
15	T-Spine 70%	-	-	423.83	202	423.83	7.54	Vertebral body T7,8, herniated disc
16	T-Spine 70%	-	-	748.1259	162	748.1259	13.32	Vertebral body T9, T10, disc herniation, spinal cord

**Table 3 brainsci-11-00646-t003:** Scan length and DLP of scout and control scan, and total effective dose with fused image datasets with iCT registration and visualized objects in augmented reality.

Number	Control Scan	Protocol	Scout Scan DLP (mGy.cm)	Scout Scan Length (mm)	iCT Scan DLP (mGy.cm)	iCT Scan Length (mm)	Total DLP (mGy.cm)	Total ED (mSv)
1	-	-	-	-	-	-	-	-
2	-	-	-	-	-	-	-	-
3	Herniated disc extent of resection control	T-Spine 30%	25.99	205	289.63	134	315.62000	5.62
4	-							
5	Implant control	L-spine 30%	18.20057	127.9999	438.00650	127	456.20707	9.03
6	-							
7	Implant control	L-Spine 70%	13.7	219	150.26	145	163.96000	3.25
8	-	T-Spine 70%	19.52	141	106.39	103	125.91000	2.24
9	-	-	-	-	-	-	-	-
10	-	-	-	-	-	-	-	-
11	Tumor resection and implant control	T-Spine 70%	23.56	181	161.31	136	184.87000	3.29
12	Implant control	T-Spine 70%	23.66	182	180.32	141	203.98000	3.63
13	-	-	-	-	-	-	-	-
14	-	-	-	-	-	-	-	-
15	Herniated disc extent of resection control	T-Spine	-	-	935.32	133	935.32	16.65
16	-	-	-	-	-	-	-	-

## Data Availability

The data presented in this study are available on request from the corresponding author. The data are not publicly available due to privacy restrictions.

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
