# Peer review of "Intraoperative Computed Tomography-Based Navigation with Augmented Reality for Lateral Approaches to the Spine"

_brainsci, 2021, doi:10.3390/brainsci11050646_

Round 1

Reviewer 1 Report

This is a very nice study on the use of intraoperative CT for navigation in lateral position and describes the advantages that navigation provides as well as the important issue of patient and staff radiation exposure. AR has been used in 10/16 cases.

The sample size is small compared to other published studies. Also, there are multiple pathologies included making the small sample size non-homogenous.

The manuscript does not state the total number of lateral surgeries performed at the study center during the study period. It would be interesting to know the ratio between navigated and non-navigated cases and when iCT was used and why. Which were the inclusion criteria in this study?

How was the placement of the reference frame decided? What is the average distance from reference frame to the surgical site? Is the use of the rector arm as accurate as the iliac crest?

The manuscript states fusion of iCT images with preoperative images (CT and MRI). Is the quality of iCT alone good enough for bony navigation or is fusion with preop CT mandatory?

Did the authors measure the accuracy of the AR-overlay? Or is this verified visually in the microscope?

Tables 2 and 3 are a bit confusing. What is the difference? Is table 2 the initial planning scan and table 3 the control scan when performed? In that case, what is the total patient exposure for the whole procedure in each case?

Why was AR used in only 10 cases? Is there any objective measure on the value of AR in these cases available? For instance, comparison to non-navigated retrospective cases?

The figure legend for figure 1 does not match the figure. The are no A1-3, B1-3, C1-4, D1-2 or E1-2. There is only letters A-N.

Figure 3, the color of the sequestered disc is changing between red and blue. Could the authors provide a video to offer a better understanding of the use of AR?

Did the authors revise or adjust anything based on the information obtained in the control iCT scans?

This is a very good study on the use of iCT navigation. The discussion is centered around the undisputed benefits of navigation using iCT and radiation doses. The role of AR is not discussed and the benefits are unclear.

The authors have previously published this method in several pioneering studies (ref 14-17). What is the novelty of the current manuscript? The lack of a control group is also a limitation of the study.

Reviewer 2 Report

The text is engaging and may be of use to a practitioner. The introduction and discussion are well written. However, the results are questionable and need to be corrected.

The novelty of the paper is the report on iCT guided spinal navigation using augmented reality for the lateral approach to oncological and infectious diseases of the spine. However, reporting results, the authors do not point to the benefits for the surgeon (crucial point) that resulted from the image-guided surgery techniques used during surgical procedures. A concise description of such benefits for each case is necessary, especially for cases of oncological and infectious diseases of the spine.  It should be a crucial part of the results. Detailed technical comments on each case are essential. The illustrative material is excellent, but it should complement your detailed description of the results.

Cases should be grouped in terms of disease entities that are an indication for the procedure. The order of figures should result from this arrangement.

Was neuromonitoring applied during the procedures?

Numerous stylistic, editorial, and linguistic corrections are necessary.

Tables should be carefully formatted.

All figures must be of high resolution. I suggest disabling compression when saving the file.

I suggest standardizing the description of the figures. For better readability, letter designations of individual images should be placed in parentheses in the figures' captions, e.g. (A) instead of A.

Figure 1 – please, describe images (A)-(N)

Figure 9 – line 222 – "E. 3D rendering…" should be replaced by "(F) 3D rendering…"

Keywords should be changed as follows: augmented reality; computer-assisted surgery; effective radiation dose; image-guided surgery; intraoperative imaging; spine navigation; lateral approach to the spine.

More recent sources should be cited in the discussion.

I suggest including the following papers to references:

Zausinger S, Scheder B, Uhl E, Heigl T, Morhard D, Tonn JC. Intraoperative computed tomography with integrated navigation system in spinal stabilizations. Spine (Phila Pa 1976). 2009 Dec 15;34(26):2919-26. doi: 10.1097/BRS.0b013e3181b77b19.

Yingsakmongkol W, Wathanavasin W, Jitpakdee K, Singhatanadgige W, Limthongkul W, Kotheeranurak V. Psoas Major Muscle Volume Does Not Affect the Postoperative Thigh Symptoms in XLIF Surgery. Brain Sci. 2021 Mar 11;11(3):357. doi: 10.3390/brainsci11030357. 

Soldozy S, Montgomery SR Jr, Sarathy D, Young S, Skaff A, Desai B, Sokolowski JD, Sandhu FA, Voyadzis JM, Yağmurlu K, Buchholz AL, Shaffrey ME, Syed HR. Diagnostic, Surgical, and Technical Considerations for Lumbar Interbody Fusion in Patients with Osteopenia and Osteoporosis: A Systematic Review. Brain Sci. 2021 Feb 14;11(2):241. doi: 10.3390/brainsci11020241.

Round 2

Reviewer 1 Report

The authors have revised the manuscript in a satisfactory manner adding additional information and correcting minor errors. 

I have no further objections.

Reviewer 2 Report

In the revised version of the manuscript, the authors applied all the corrections suggested in my review. The manuscript in its current form does not raise any objections. Figures are excellent, and the descriptions of the results have been significantly expanded. The decision to use the MPDI English editing service for stylistic, editorial, and linguistic correction was reasoned.

Congratulations to the authors, and I appreciate their effort put into revising the manuscript. In my opinion, it is a really nice paper.